# Effectiveness of a participatory approach to develop school health interventions in four low resource cities: study protocol of the 'empowering adolescents to lead change using health data' cluster randomised controlled trial

Regina Guthold [iD],[1] Laura Kann,[2] Lubna Bhatti,[2] Parviz Abduvahobov,[3] Joana Ansong,[4] Uki Atkinson,[5] Valentina Baltag,[1] Sonja Caffe,[6] Roberta Caixeta,[7] Cheick Bady Diallo,[8] Heba Fouad,[9] Sally Haddad,[10] Hafid Hachri,[11] Jeannine A Jaggi,[10] Pradeep Joshi,[12] Priya Karna,[12] Sidi Allal Louazani,[2] Symplice Mbola Mbassi,[13] Rajesh Mehta,[14] Yogendra Mudgal,[15] Claudio R Nigg [iD],[10] Anthony D Okely,[16] Dolores Ondarsuhu [iD],[7] Tahar Ouaourir,[17] Fatima Zahra Trhari,[17] Leanne M Riley[2]

**Correspondence to**
Dr Regina Guthold;
gutholdr@who.int

## ABSTRACT

**Introduction** Comprehensive local data on adolescent health are often lacking, particularly in lower resource settings. Furthermore, there are knowledge gaps around which interventions are effective to support healthy behaviours. This study generates health information for students from cities in four middle-income countries to plan, implement and subsequently evaluate a package of interventions to improve health outcomes.

**Methods and analysis** We will conduct a cluster randomised controlled trial in schools in Fez, Morocco; Jaipur, India; Saint Catherine Parish, Jamaica; and Sekondi-Takoradi, Ghana. In each city, approximately 30 schools will be randomly selected and assigned to the control or intervention arm. Baseline data collection includes three components. First, a Global School Health Policies and Practices Survey (G-SHPPS) to be completed by principals of all selected schools. Second, a Global School-based Student Health Survey (GSHS) to be administered to a target sample of n=3153 13–17 years old students of randomly selected classes of these schools, including questions on alcohol, tobacco and drug use, diet, hygiene, mental health, physical activity, protective factors, sexual behaviours, violence and injury. Third, a study validating the GSHS physical activity questions against wrist-worn accelerometry in one randomly selected class in each control school (n approximately 300 students per city). Intervention schools will develop a suite of interventions using a participatory approach driven by students and involving parents/guardians, teachers and community stakeholders. Interventions will aim to change existing structures and policies at schools to positively influence students' behaviour, using the collected data and guided by the framework for *Making Every School a Health Promoting School*. Outcomes will be assessed for differential change after a 2-year follow-up.

## STRENGTHS AND LIMITATIONS OF THIS STUDY

⇒ This study generates comprehensive individual-level and school-level adolescent health information for direct use to plan, implement and subsequently evaluate a package of interventions to improve health outcomes across four cities from different world regions, with a focus on underserved locations.
⇒ The approach to develop interventions is data-informed, systematic and evidence-based, culturally adapted and flexible and participatory, driven by adolescents, parents and school and community stakeholders.
⇒ This study makes use of existing global tools for data collection, intervention planning and implementation and builds on local policies and practices as a foundation for action.
⇒ The COVID-19 pandemic and related school closures have led to delays in study implementation.
⇒ Internet connectivity may be poor in some study sites, and alternative methods for study components with electronic data collection may need to be explored.

**Ethics and dissemination** The study was approved by WHO's Research Ethics Review Committee; by the Jodhpur School of Public Health's Institutional Review Board for Jaipur, India; by the Noguchi Memorial Institute for Medical Research Institutional Review Board for Sekondi-Takoradi, Ghana; by the Ministry of Health and Wellness' Advisory Panel on Ethics and Medico-Legal Affairs for St Catherine Parish, Jamaica, and by the Comité d'éthique pour la recherche biomédicale of the Université Mohammed V

of Rabat for Fez, Morocco. Findings will be shared through open access publications and conferences.

**Trial registration number** NCT04963426.

## INTRODUCTION

Adolescence is a unique period in life with rapid physical, cognitive and psychosocial growth. It is an important time for laying foundations of future health, including, for example, establishing behaviours related to noncommunicable diseases (NCDs) such as tobacco and alcohol use, diet and physical activity, as well as behaviours linked to sexual or mental health.[1] Implementing effective, evidence-based interventions could substantially reduce risk behaviours and support protective factors among adolescents and improve their future adult health.[2 3] However, several critical barriers exist to implementing such interventions for adolescents.

First, there is a lack of relevant and context-specific adolescent health information, particularly in lower resource settings, that is essential to inform intervention planning.[4] Where data exist, they are primarily national-level estimates on specific health conditions that often mask variations and unequal progress by smaller geographical units.[5 6] Local data are needed to uncover and address these inequalities and develop the most relevant, effective, age-appropriate and context-specific interventions for adolescents. This includes data on policies and practices affecting adolescent's behaviour, and on their health behaviours and protective factors. Collecting this information is of particular importance in usually neglected urban settings where little data exist, such as cities other than capitals (called 'secondary cities' hereafter).

Second, information on adolescent health is sometimes collected with questionnaires that may not have been validated for the populations in which they are used. Physical activity is one example where the most internationally used questions for adolescents have only been tested for validity in a few high-income countries.[7 8] To produce useful data to inform decision-making, it is essential for these questions to be tested more widely. Another reason to test adolescent physical activity questions is related to the change in recommended levels of physical activity for this age group. Most currently used questions are based on the 2010 WHO guidelines on physical activity that recommended adolescents to be physically active for at least 60 min *each* day.[9] In 2020, these guidelines were updated and now state that adolescents should engage in at least *an average* of 60 min of activity per day throughout the week.[10] Consequently, there is a need for adolescent physical activity questions to be tested against the updated guidelines.

Third, there are significant knowledge gaps about which interventions might be effective in targeting risk behaviours and supporting protective factors in adolescents.[11] Available evidence is strongest for universal school-based interventions targeting multiple risk behaviours, however, most of the included studies were conducted in high-income countries, while evidence from lower resource settings is lacking.[11]

Fourth, adolescents are too rarely involved in health promotion planning, decision-making and implementation. Yet, participatory approaches engaging adolescents are crucial for sustainable change in cognitions and behaviour[12 13] and thus warranted.

Global tools that help overcome the aforementioned barriers exist. The Global School Health Policies and Practices Survey (G-SHPPS) generates important school-level policy information.[14] The Global School-based Student Health Survey (GSHS) collects data on students' health behaviours and protective factors.[15] The Global Accelerated Action for the Health of Adolescents (Global AA-HA!) provides a systematic, participatory approach to understanding adolescent health needs based on data, prioritising these needs in the local context and planning, implementing, monitoring and evaluating appropriate interventions.[16] The initiative *Making Every School a Health Promoting School* provides an evidence-based framework for the implementation of eight Global Standards to improve students' health[17] and promotes a whole-school approach that has demonstrated positive effects on health.[18–21] Both the Global AA-HA! and *Making Every School a Health Promoting School* emphasise the importance of students' involvement in health promotion planning, decision-making and implementation in a manner that is empowering. Finally, photovoice facilitates adolescent engagement by allowing them to pictorially take impressions of health facilitators and barriers in their schools and community to inform intervention planning.[22–24]

These available global tools will be used to achieve the goals of this study. The overarching aim is to generate health information for students of secondary cities from four middle-income countries of different world regions that will be directly used to plan, implement and evaluate a package of interventions for programme planning, policy and structural change to improve students' health.

Specific objectives include: First, to assess current health policies and practices in selected schools of the four cities. Second, to understand the levels of health risk and protective factors among school-going adolescents in these cities. Third, to validate currently used questions to assess students' physical activity behaviour with wearable accelerometers. Fourth, to plan and evaluate a participatory intervention approach that focuses on changing existing structures, policies and practices and on implementing programmes in and around schools to improve health outcomes for students.

## METHODS AND ANALYSIS
### Study design and locations

We will conduct a multisite two-armed cluster randomised controlled trial using a nested cross-sectional design. The clusters are defined as schools and classes within schools. The study locations are spread across four world regions

and include the cities of Fez, Morocco; Jaipur, India; Saint Catherine Parish, Jamaica; and Sekondi-Takoradi, Ghana. These locations (secondary cities) were purposefully selected in collaboration with WHO regional focal points to generate information for settings that are less likely to be in the focus or receive resources for adolescent health interventions.

In each city, approximately 30 schools will be selected and randomly assigned to either the intervention or control group. At baseline, all sampled schools will conduct a G-SHPPS to assess school policies and practices[14] and a GSHS to assess risk behaviours and protective factors among students.[15] Additionally, in the control group of each city, we will conduct studies to validate GSHS physical activity questions against an accelerometer (Axivity AX3[25]) to be worn by students on the wrist for 8 days prior to the baseline survey.

Following the baseline surveys, the intervention schools will participate in data-to-action workshops to develop a package of interventions. Students, parents, teachers, local authorities and researchers will work together to find and prioritise policy and programme solutions to establish a unified action plan. Subsequently, these interventions will be implemented, and study processes and outcomes will be assessed for differential change via repeat G-SHPPS and GSHS surveys after a 2-year follow-up. At follow-up, new classes will be selected from the same schools to ensure participation of 13–17 years old students similar in age to the baseline sample. We hypothesise that the prevalence of behavioural risk factors would be lower and the prevalence of protective factors higher among students in schools in the intervention group compared with the control group at follow-up. An overview of our study design is shown in figure 1.

## Study population and sampling

This study's target population is 13–17 years old students. In each city, following standard G-SHPPS and GSHS sampling procedures,[14 15] approximately 30 schools will be randomly selected from a list of schools including every school in the city with eligible classes (classes that the target population usually attends), provided by education authorities from the respective cities. Selection will be done by experts from WHO with probability proportional to the number of students in the eligible schools.

For the GSHS, in a second sampling stage, classes will be randomly selected from all eligible classes. All students of selected classes will be asked to participate in the survey.

The city sample size was calculated to test the hypothesis that the prevalence of behavioural risk factors would be lower among students in the schools of the intervention group versus those in the control group at follow-up using G*Power V.3.1.[26] With most outcome variables (prevalence of behavioural risk and protective factors) being binary, logistic regression was selected as the statistical test, the required power level was set at 80% and the significance level prespecified at α=0.05. The to-be-detected effect was specified at an OR=0.75, which represents the average of pooled effect sizes from a recent meta-analysis[11] considering universal school interventions across a range of behaviours, similar to the ones assessed in our study.

The initial sample size calculation resulted in n=473, which was inflated assuming a design effect of 1.5 and

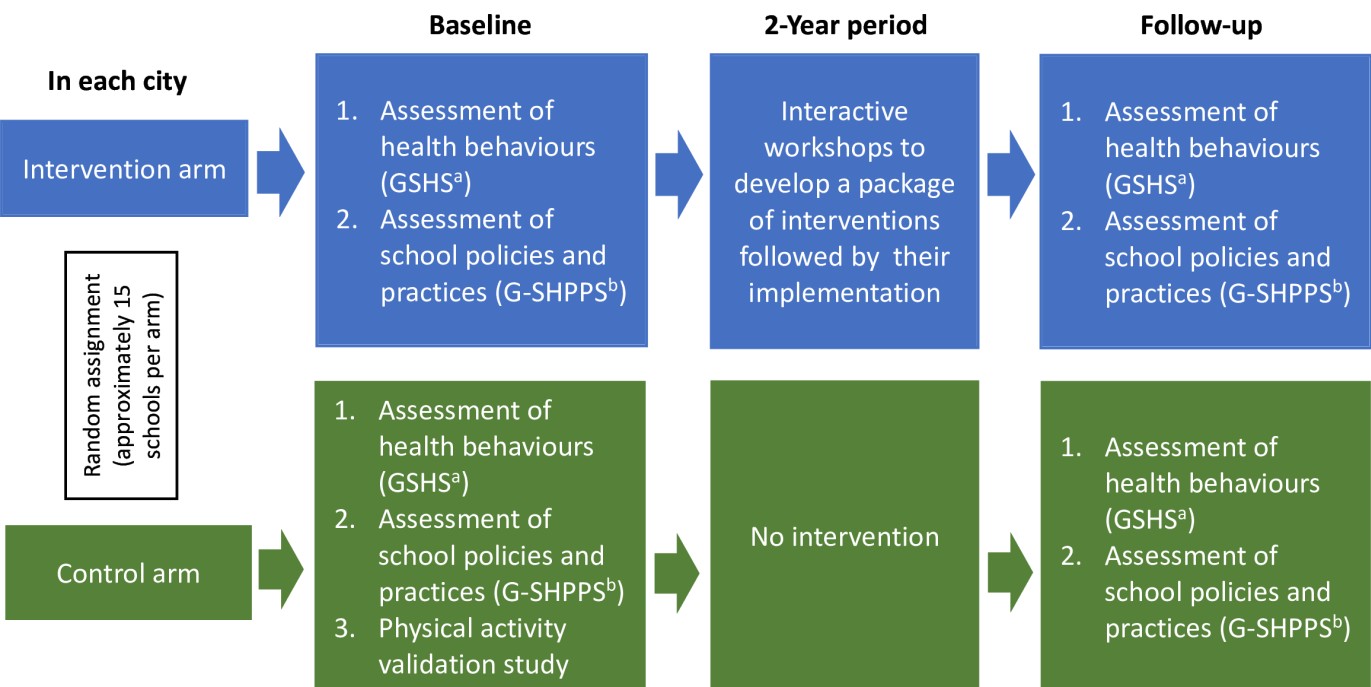

**Figure 1** Overview of the study design. [a] GSHS, Global School-based Student Health Survey; [b] G-SHPPS, Global School Health Policies and Practices Survey.

**Table 1** Questionnaire modules of the Global School Health Policies and Practices Survey (G-SHPPS) and sample process indicators

| G-SHPPS module | Sample process indicators (Percentage of schools that…) |
|---|---|
| School Health Services | ▶ Routinely provide as part of the health services offered to students counselling to prevent violence (including intimate partner, violence, sexual violence, gender-based violence, bullying, and gang violence).<br>▶ Routinely provide as part of the health services offered to students administration of recommended immunizations.<br>▶ Routinely provide as part of the health services offered to students support for management of overweight and obesity. |
| School Physical environment | ▶ Provide soap and water at handwashing facilities for students.<br>▶ Have improved sanitation facilities for students. |
| Food and Nutrition Services | ▶ Make sugar-sweetened carbonated soft drinks and other sugar-sweetened drinks available to students through the vending machines, stores, canteens, or snack bars on school premises.<br>▶ Routinely make fruits and vegetables available to students through the school's food and nutrition services.<br>▶ Price healthy foods and beverages in the vending machine, school store, canteen, or snack bar on school premises lower than unhealthy options. |
| Health Education | ▶ Teach health education.<br>▶ Teach sexual and reproductive health and HIV transmission, prevention, and treatment to students.<br>▶ Teach violence and bullying prevention to students.<br>▶ Teach decision-making, problem-solving, goal-setting, and refusal skills to students to help them avoid or reduce health risks. |
| Physical education | ▶ Teach physical education.<br>▶ Teach students as part of physical education the value of physical activity for health, enjoyment, challenge, self-expression, and/or social interaction.<br>▶ Regularly provide recess or other physical activity breaks to students during the school day. |
| School Governance and Leadership | ▶ Have an official council, committee, or team responsible for implementing health promoting school policies, programmes, and activities.<br>▶ Regularly involve and engage parents/caregivers/families and students in developing health promoting school policies. |
| School Policies and Resources | ▶ Have a policy specifically about becoming or continuing to be a Health Promoting School.<br>▶ Have a policy prohibiting use of at least some tobacco or nicotine products, alcohol, and Illicit drugs among students on school premises.<br>▶ Regularly monitor, evaluate, or assess the quality of their health promoting policies, programmes, or activities. |

an expected response rate of 90%: 473×1.5/0.9=788. Finally, this sample size was quadrupled aiming at reporting for two age groups within both the intervention and control arm: 13–15 and 16–17 years, resulting in a target sample size of n=3153.

Participation in the G-SHPPS and GSHS is voluntary. For the GSHS, in line with local policies, consent will be obtained from the students themselves and/or their parents/guardians (see online supplemental appendix for consent forms).

### Data collection and measures
#### G-SHPPS and GSHS
The G-SHPPS questionnaire (2021 version) will assess school policies and practices[14] across seven broad topics (table 1 and online supplemental appendix). Exemplary process indicators are presented under the topics in table 1. School principals of the participating schools will be asked to electronically complete the questionnaire.

Students' health behaviours and protective factors will be assessed using the GSHS questionnaire (2021 version)[15] (see online supplemental appendix). The GSHS is a paper-based, self-administered questionnaire with generic answer sheets that students complete during one classroom period.

The GSHS includes 10 core questionnaire modules. Table 2 presents these modules with exemplary primary outcome indicators. Each city will be encouraged to include all modules. However, up to four modules may be excluded if questions of a specific module are deemed too sensitive. Cities may add context-specific questions, not exceeding a total of 80 questions.

**Table 2** Questionnaire modules of the Global School-based Student Health Survey (GSHS) and sample primary outcome indicators

| GSHS module | Sample outcome indicators, dependent on the city's choice of modules (percentage of students who…) |
|---|---|
| Alcohol use | ▶ Currently drank alcohol (at least one drink of alcohol on at least 1 day during the 30 days before the survey). <br> ▶ Ever drank so much alcohol that they were really drunk one or more times during their life. <br> ▶ Had their first drink alcohol other than a few sips before age 14 years, among students who ever had a drink of alcohol other than a few sips. |
| Dietary behaviours | ▶ Were underweight (<−2 SD from median for BMI by age and sex) (measured). <br> ▶ Were overweight (>+1 SD from median for BMI by age and sex) (measured). <br> ▶ Were obese (>+2 SD from median for BMI by age and sex) (measured). <br> ▶ Usually drank carbonated soft drinks one or more times per day during the 30 days before the survey. |
| Drug use | ▶ Ever used cannabis one or more times during their life. <br> ▶ Used drugs before age 14 years for the first time, among students who ever used drugs. |
| Hygiene | ▶ Usually cleaned or brushed their teeth less than one time per day during the 30 days before the survey. <br> ▶ Never or rarely washed their hands after using the toilet or latrine during the 30 days before the survey. |
| Mental health | ▶ Seriously considered attempting suicide during the 12 months before the survey. <br> ▶ Attempted suicide one or more times during the 12 months before the survey. <br> ▶ Did not have any close friends. |
| Physical activity | ▶ Were physically active for a total of at least 60 min per day on all 7 days during the 7 days before the survey. <br> ▶ Attended physical education classes on 3 or more days each week during this school year. <br> ▶ Spent 3 or more hours per day during a typical or usual day sitting or lying down (doing such things as watching television, playing computer games, talking with friends, using their mobile phone, travelling in a motor vehicle, napping or doing other activities sitting or lying down) when not in school or doing homework or sleeping at night. |
| Protective factors | ▶ Missed classes or school without permission on 1 or more days during the 30 days before the survey. <br> ▶ Reported that their parents or guardians most of the time or always understood their problems and worries during the 30 days before the survey. <br> ▶ Reported that their parents or guardians most of the time or always really knew what they were doing with their free time during the 30 days before the survey. |
| Sexual behaviours | ▶ Ever had sexual intercourse. <br> ▶ Had sexual intercourse before age 14 years for the first time, among students who ever had sexual intercourse. <br> ▶ Used a condom during last sexual intercourse, among students who ever had sexual intercourse. |
| Tobacco use | ▶ Currently used any tobacco products (used any tobacco products on at least 1 day during the 30 days before the survey). <br> ▶ Currently smoked cigarettes (smoked cigarettes on at least 1 day during the 30 days before the survey). <br> ▶ Currently used electronic cigarettes (used e-cigarettes on at least 1 day during the 30 days before the survey). |
| Violence and unintentional injury | ▶ Were in a physical fight one or more times during the 12 months before the survey. <br> ▶ Were seriously injured one or more times during the 12 months before the survey. <br> ▶ Were bullied during the 12 months before the survey. |

BMI, body mass index.

Anthropometric measurements of height and weight are taken prior to survey administration. Students receive their measurement results and are asked to subsequently record them on the questionnaire.

Inclusion of the physical activity module will be mandatory to allow for validity testing of some of these questions. Additionally, a new question will be included for validation against the updated WHO physical activity guidelines[10] reading 'During the past 7 days, did you do at least an average of 60 minutes per day of physical activity across the 7 days?'.

Participating cities will also be encouraged to include the standard GSHS questions on attending school from home during the pandemic, missing classes or school without permission and on school attendance under the influence of alcohol and of drugs.

Once the questionnaire has been finalised for each city, the same questionnaire will be anonymously self-administered by all participating students.

### Additional measures
With the interventions of the present study aiming at improving students' health and including programme planning, policy and structural change, it is hypothesised that a range of secondary outcomes would also be positively influenced. These secondary outcomes will

depend on the topics and interventions selected by each city during the data-to-action workshop and will be identified on finalisation of each city's action plan. They may include school absenteeism and performance among learners in the intervention schools, improved healthcare seeking, secondary health outcomes and improved communication of adolescents with teachers and parents.

## Physical activity validation study

In addition to the GSHS self-reported physical activity, activity data will be collected using the Axivity AX3 accelerometer. This device has previously been used in large-scale population-based[27 28] and school-based studies.[29] The Axivity AX3 records movement, and reports raw acceleration signals in real time, allowing for transparent analysis.[30]

From each school in the control group, one class will be randomly selected to participate in testing the validity of questions currently used in the GSHS to assess student's physical activity, sedentary behaviour and sleep, and the new question that was added to assess WHO's 2020 physical activity guidelines.[10] Every student in the selected classes will be asked to participate (approximately 300 students per city).

Participation in the physical activity validation study is voluntary. Participating students will be asked to provide written assent, and their parents or guardians will be asked to provide written consent (see online supplemental appendix).

In the week prior to the GSHS baseline survey, selected students will wear the Axivity AX3 on their non-dominant wrist continuously for 8 consecutive days.[31] Trained personnel will instruct students to wear the waterproof device and to participate in all their normal activities. After having worn the accelerometer, students will be asked to participate in the GSHS survey at baseline. Axivity AX3 data will then be compared with data collected with the GSHS physical activity module and the added physical activity question.

## Interventions

This study's intervention approach is embedded in the framework of the Global Standards for health promoting schools and its concept of a whole-school approach to promoting health.[18] Schools in the intervention arm will implement health promotion strategies to address the most critical health risk behaviours and protective factors, considering the eight Global Standards including government policies and resources; school policies and resources; school governance and leadership; school and community partnerships; school curriculum; school social-emotional environment; school physical environment; and school health services.

Following the baseline surveys, students in the intervention arm will receive instructions to participate in photovoice.[22] This is an established[23 24] qualitative participatory method through which adolescents can pictorially take impressions of health facilitators and barriers in their school and community. Pictures will be uploaded onto a secure platform, reviewed and sorted by trained personnel, ensuring that no inappropriate photos are included.

To develop the interventions, data-to-action workshops will be held in each city (see online supplemental appendix for agenda). They will be facilitated by the local study coordinator and WHO and UNESCO personnel and include a minimum of three nominated students from each intervention school, teachers, parents and up to five local authorities and researchers. The nomination of students for the workshops will be based on the student's interest in participating and ensure gender balance.

Data-to-action workshops will use the data collected during the baseline surveys, the photovoice pictures and follow the Global AA-HA! approach and its three key steps[16]: First, recognising that the health needs are different in each city, a needs assessment will be done, using the GSHS data and the photovoice pictures. To facilitate this, for the workshop, data will be visualised in an easy-to-understand summary format, including fact sheets and infographics. Workshop participants will review the site-specific results to identify which conditions, health risks and determinants need to be addressed most urgently. Second, considering that the situation adolescents live in is different in each setting, in a landscape analysis, results from the local G-SHPPS will be reviewed, along with other existing programmes, policies and legislation. Similar to step one, results will be summarised in an easy-to-understand format and used to identify gaps and potential areas for improvement. Step two will also include a review of evidence-based interventions of the Global AA-HA! and a meta-analysis,[11] prepared in a format that will speak to workshop participants. Third, specific actions will be identified to address the issues specific to each city, considering the outcomes of steps one and two. Identified actions will be prioritised, considering availability of resources, effectiveness, feasibility, capacity to implement, cultural appropriateness and focus on structural and systems-oriented changes.

This participatory approach will ensure that—while considering the adolescents' health needs and each city's situation—the selected interventions will be based on evidence, be feasible, appropriate, acceptable and aligned with local priorities. For example, a set of interventions to decrease tobacco use might be selected for a city with a high prevalence of tobacco use among students. These might include individual-level and interpersonal-level interventions, but also school policy changes such as the introduction of smoke-free school policies that would ultimately lead to sustained behaviour change.

The workshop duration will be 3 days, and include students, parents and teachers for at least 2 hours per day. To ensure that adolescents are meaningfully engaged, existing guidelines on adolescent participation and civic engagement will be followed.[32] Breakout sessions will be organised during the workshops for which the different stakeholders will be separated, to ensure that each

stakeholder group gets their own space to express their views. The different views will be brought back to the plenary and integrated as appropriate. Students will also get the opportunity to comment on and add to actions proposed by other stakeholders.

Each city will develop a 2-year implementation plan based on the prioritised interventions. These will be classified by topic and implementation level, including structural, environmental, organisational, community, interpersonal and individual level. A team of local stakeholders will be identified with clear responsibilities regarding implementation of the interventions, and additional technical support will be provided by WHO.

### Monitoring and evaluation

As per the Global AA-HA!,[16] each city's implementation plan will include process and outcome monitoring through data collected with the G-SHPPS and the GSHS. Progress in the implementation of interventions will be monitored by the team of local stakeholders and WHO on a periodic basis. For each agreed on activity/intervention in the implementation plan, monthly reporting will include whether the activity is not started, in planning, in progress or implemented, with a section for next steps and who is responsible.

Schools in the control arm will operate under 'business as usual' within their specific context, that is, they will not receive any intervention selected during the data-to-action workshop.

Follow-up G-SHPPS and GSHS surveys (the same as at baseline) will occur 2 years after the baseline surveys in all selected schools. For the GSHS, new classes from these schools will be randomly selected to ensure participation of 13–17 years old students similar in age to the baseline sample.

### Data management and analysis plan

During electronic data collection through Dataform (LimeSurvey),[33] G-SHPPS data are uploaded to a secure server that will only be accessible by the city and WHO. Summary results of key indicators will be produced for each city.

GSHS data are collected on computer scannable answer sheets with no personal identifying information. Data will be processed using standard cleaning and generic data analysis programmes. A city-specific database will be generated and results will be displayed on a standard fact sheet including information on response rates, a summary of methods and weighted prevalence estimates with CIs.[15]

The Axivity AX3 data will be downloaded on-site and uploaded to a secure WHO server with unique identifiers, with no personal information. The unique identifier will be used to link these data to the student's GSHS physical activity data. Time spent in total physical activity, moderate-to-vigorous-intensity physical activity, and in sedentary behaviour will be computed from the Axivity AX3 data for each student, using standard methods.[34 35]

Agreement between the Axivity AX3 and the questionnaire data will be assessed using approaches consistent with previous studies.[31 36]

For GSHS and G-SHPPS follow-up surveys, the same data analysis procedures as at baseline will be used.

To assess effectiveness of the interventions, comparative analysis of changes of policies and programmes and of students' health behaviours between the intervention and control group will be undertaken. With most outcome variables being binary, most intervention effects will be estimated through adjusted ORs with 95% CIs, using random effects logistic regression to adjust for within-school clustering. Additionally, a score across multiple risk and protective behaviours will be computed for each city (depending on the selection of questionnaire modules in each city), and effectiveness of the intervention will also be tested using this score.[37] Data analysts will be blinded to allocation of schools in the two arms.

The analysis will account for potential spillover effects as much as possible. Spillover effects are defined as benefits provided by interventions that extend beyond direct recipients and impact people in close physical or social proximity who did not directly receive the intervention themselves, namely students in the control arm.[38]

### Patient and public involvement statement

Relevant local stakeholders will be engaged at all project stages to ensure that this research builds local capacity and is responsive to the health needs and priorities of the students in each city.[39] Besides representatives of the national or local health and education authorities and local researchers, the study committees include community and school stakeholders. Student participants, parents/guardians, teachers and community leaders will lead the use of the collected survey data to identify and implement interventions. Interventions will additionally be shaped by pictures that students provide of what they perceive as facilitators and barriers to their own health behaviour.

### Trial status

This trial began recruitment in January 2022. Baseline data collection started in the first city (Jaipur; India) in August 2022 and has been finalised in all four cities in May 2023. After the 2-year intervention period and the follow-up assessments, the study completion date is expected in January 2026. WHO as the trial management body will provide oversight to this multicity trial throughout its duration.

## ETHICS AND DISSEMINATION

The master protocol and the four site-specific protocols have been approved by the WHO Research Ethics Review Committee (ERC.0003397). The four site-specific protocols have also been approved by the Jodhpur School of Public Health's Institutional Review Board for Jaipur, India; the Noguchi Memorial Institute for

Medical Research Institutional Review Board for Sekondi-Takoradi, Ghana; the Ministry of Health and Wellness' Advisory Panel on Ethics and Medico-Legal Affairs for St Catherine, Jamaica, and the Comité d'éthique pour la recherche biomédicale of the Université Mohammed V of Rabat for Fez, Morocco. Following standard G-SHPPS procedures, permissions from all schools to take part in the study will be obtained prior to field work. For the GSHS, following standard procedures previously used in over 100 countries, all parents/guardians will be notified about the study and parental/guardian consent will be obtained following local laws and policies. For the physical activity validation study, all parents/guardians will be notified and parental/guardian consent and/or student assent will be obtained in each city, adhering to local laws and policies.

The present study is minimally intrusive on the student's privacy, on schools and communities. All data collection tools have previously been tested and used in populations similar to the target population of this study. Through participatory planning and careful review, it will be ensured that the interventions are of minimal physical, psychological and social risk. It is expected that the target population will benefit from the outcome of the interventions, in the form of improved policies and practices, as well as improved behavioural outcomes. In each city, local stakeholders will determine appropriate compensation for participating students' time spent at the data-to-action workshop and to co-design interventions.

Following GSHS and G-SHPPS policy, data will be held at the country level, with a copy at WHO, for an initial 2-year period. During this time, each site will be encouraged to produce any report and publication they desire. After this period, data sets will be made publicly available, including in the WHO NCD microdata repository.[40] GSHS and G-SHPPS results will be presented in a summary report, including user-friendly infographics, for sharing with the participating students, their parents, teachers and communities. Data and results from the physical activity validation study will be stored on a WHO server and will be made available to interested researchers on request after the initial 2-year period.

## Author affiliations
[1]Maternal, Newborn, Child and Adolescent Health and Ageing Department, WHO, Geneva, Switzerland
[2]Noncommunicable Diseases Department, WHO, Geneva, Switzerland
[3]Health and Education Section, Division for Peace and Sustainable Development, Education Sector, UNESCO, Paris, France
[4]WHO Country Office for Ghana, Accra, Ghana
[5]National Council on Drug Abuse, Kingston, Jamaica
[6]Family, Health Promotion and Life Course, PAHO, Washington, Columbia, USA
[7]Noncommunicable Diseases and Mental Health Department, PAHO, Washington, Columbia, USA
[8]Universal Health Coverage/Communicable and Noncommunicable Diseases, WHO Regional Office for Africa, Brazzaville, Congo
[9]Noncommunicable Diseases and Mental Health Department, WHO Regional Office for the Eastern Mediterranean, Cairo, Egypt
[10]Department of Health Science, Institute of Sport Science, University of Bern, Bern, Switzerland
[11]WHO Country Office for Morocco, Rabat, Morocco
[12]WHO Country Office for India, New Delhi, India
[13]Universal Health Coverage/Life Course, WHO Regional Office for Africa, Brazzaville, Congo
[14]WHO Regional Office for South-East Asia, New Delhi, India
[15]Office of Joint Director, School Education, Jaipur, India
[16]School of Health and Society, University of Wollongong, Wollongong, New South Wales, Australia
[17]Population Department, Ministry of Health, Rabat, Morocco

**Acknowledgements** We thank Professor Susan Sawyer and Professor Antony Morgan for their thorough review of the study protocol. We also gratefully acknowledge Professor Fiona Bull's contribution to the study protocol and advice on the physcial activtiy component of the study.

**Contributors** RG wrote the first draft of this paper. RG and LMR conceptualised the study and developed the methods. LK, LB and SAL contributed to the development of the methods and preparation for sampling and data collection. PA contributed to the development of table 2 and the methods for the intervention. ADO contributed to the development of the methods for the physical activity validation study. CRN, SH, JAJ and VB contributed to the development of the methods for the intervention. JA, UA, SC, RC, CBD, HF, HH, PJ, PK, SMM, RM, YM, DO, TO and FZT coordinate site-specific implementation and provided input on the manuscript from a site-specific perspective. All authors have reviewed the paper and approved its final version.

**Funding** This work is supported by Fondation Botnar, Activity REG-19-015. The funder did not play a role in the design of this study.

**Disclaimer** The author is a staff member of the World Health Organization. The author alone is responsible for the views expressed in this publication and they do not necessarily represent the views, decisions or policies of the World Health Organization.

**Competing interests** None declared.

**Patient and public involvement** Patients and/or the public were involved in the design, or conduct, or reporting, or dissemination plans of this research. Refer to the Methods section for further details.

**Patient consent for publication** Not applicable.

**Provenance and peer review** Not commissioned; externally peer reviewed.

**Author note** RG, LK, LB, JA, VB, CBD, HF, HH, PJ, PK, SAL, SMM, RM and LMR are members of the WHO. SC, RC and DO are members of the Pan American Health Organization. The authors alone are responsible for the views expressed in this publication and they do not necessarily represent the decisions, policy or views of the WHO or the Pan American Health Organization.

**ORCID iDs**
Regina Guthold http://orcid.org/0000-0003-3073-6468
Claudio R Nigg http://orcid.org/0000-0002-2897-4689
Dolores Ondarsuhu http://orcid.org/0000-0003-0506-3046

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
