## [Reviewer comments · BMJ Open]

ARTICLE DETAILS

TITLE (PROVISIONAL)	Effectiveness of a participatory approach to develop school health interventions in four low resource cities: Study protocol of the “Empowering adolescents to lead change using health data” cluster randomized controlled trial
AUTHORS	Guthold, Regina; Kann, Laura; Bhatti, Lubna; Abduvahobov, Parviz; Ansong, Joana; Atkinson, Uki; Baltag, Valentina; Caffè, Sonja; Caixeta, Roberta; Diallo, Cheick Bady; Fouad, Heba; Haddad, Sally; Hachri, Hafid; Jaggi, Jeannine; Joshi, Pradeep; Karna, Priya; Louazani, Sidi Allal; Mbola Mbassi, Sympllice; Mehta, Rajesh; Mudgal, Yogendra; Nigg, Claudio; Okely, Tony; Ondarsuhu, Dolores; Ouaourir, Tahar; Trhari, Fatima Zahra; Riley, Leanne

VERSION 1 – REVIEW

REVIEWER	Cristi-Montero, Carlos Pontificia Universidad Católica de Valparaíso, IRyS Group, Physical Education School
REVIEW RETURNED	02-Apr-2023

GENERAL COMMENTS	Dear Editor, thanks for the invitation to review this protocol study. The authors present a novel and relevant protocol covering a critical research gap, raising valuable information for future health programs. Here are some general comments about this work: Strengths The protocol has been registered. The study design, hypothesis, and methodology are well-focused to reach the aims proposed. The study involves low- and middle-income countries from several countries. Data collection will be at the individual and school level. The project includes school principals as students getting a wide landscape to establish future interventions or programs. Currently, the last group is poorly involved in health promotion planning. Besides, it includes a study validating the GSHS physical activity questions—an outstanding contribution to this matter. Sample size calculation is declared. Questions 174/page 12. The project involves four world regions, but why these countries and not others? For example, why not any from Latin America?
--

	361/page 22. They indicate that no imputation for missing data will be performed. It could be helpful if you indicate why not. Suggestion Primary outcomes are well declared; however, secondary outcomes/analyses could emerge in the future, considering the valuable data that will be collected. It could be helpful if secondary analyses are declared.
--	--

REVIEWER	Hamdani, Syed Usman Global Institute of Human Development, Public Mental Health
REVIEW RETURNED	19-May-2023

GENERAL COMMENTS	This is a significant research area and a crucial step toward optimizing adolescents' health in low-resource settings. I really appreciate the team of authors for prioritizing one of the public health concerns in such a comprehensive way. The research seems to be meticulously planned and scientifically well-written; specifically, the involvement of adolescents and co-designing interventions with youth involvement is an important area. A few concerns can be addressed to strengthen the manuscript further. Specific Comments:  - Randomization and blinding: Please elaborate on who will generate the allocation sequence for all sites included in the multi-country trial. Details on blinding need to be added (who will be blind to the allocation status of the participants). - Intervention: The authors have mentioned that the intervention will be co-designed and components of the proposed interventions will be prioritized after the needs assessment phase; the description of the intervention lacks necessary information, e.g., duration of the intervention, who will be the delivery agents, how fidelity of the intervention across sites will be ensured, etc. - Control condition: The description of control arms needs to be added. Considering it is a multi-site trial, treatment as usual for adolescents attending schools across all sites may depend on the context. A brief description of the control condition could facilitate seeing how the intervention will compare across sites. - Primary and secondary outcomes need to be specified as outlined in the trial registry. - Participant's two-year follow-up: A brief description of the schools enrolled in the trial needs to be added, e.g., whether these are 10th or 12th-grade schools. Realizing all those participants who will be aged 17 at the time of enrolment will turn 19 years old at the time of primary end-point follow-up and might join the college by the time follow-up assessment will be conducted? A brief description of how all those participants who might have left the schools during the study will be followed up at the time of the primary outcome needs to be added. - Participants' safety: Please address any potential risks or benefits to participants and how they will be mitigated or maximized, respectively, during their participation in the trial across sites. Also, please explain whether participants, especially young adolescents, will be compensated for their time spent co-designing workshops. - Trial governance: Please add a brief description of the trial government or trial management body that will provide oversight to a multi-site trial throughout the duration of the trial.
--

VERSION 1 – AUTHOR RESPONSE

Reviewer #1	
Strengths The protocol has been registered. The study design, hypothesis, and methodology are well-focused to reach the aims proposed. The study involves low- and middle-income countries from several countries. Data collection will be at the individual and school level. The project includes school principals as students getting a wide landscape to establish future interventions or programs. Currently, the last group is poorly involved in health promotion planning. Besides, it includes a study validating the GSHS physical activity questions—an outstanding contribution to this matter. Sample size calculation is declared.	Thank you for these comments. We have addressed the specific comments as per our responses below.
174/page 12. The project involves four world regions, but why these countries and not others? For example, why not any from Latin America?	Thank you for this comment. We have now added information on how and why these locations were selected.
361/page 22. They indicate that no imputation for missing data will be performed. It could be helpful if you indicate why not.	Thank you for this comment. We acknowledge that – depending on some of the data values – it may be valuable to consider data imputation at the stage of data analysis. To allow for this, we have not taken this sentence out.
Suggestion Primary outcomes are well declared; however, secondary outcomes/analyses could emerge in the future, considering the valuable data that will be collected. It could be helpful if secondary analyses are declared.	Thank you. We have added a paragraph on ‘Additional measures’ where we now describe secondary outcomes.
Reviewer #2	
General comments This is a significant research area and a crucial step toward optimizing adolescents’ health in low-resource settings. I really appreciate the team of authors for prioritizing one of the public health concerns in such a comprehensive way. The research seems to be meticulously planned and scientifically well-written; specifically, the involvement of adolescents and co-designing interventions with youth involvement is an important area. A few concerns can be addressed to strengthen the manuscript further.	Thank you very much for these comments. We have addressed the specific comments as per our responses below.

Randomization and blinding: Please elaborate on who will generate the allocation sequence for all sites included in the multi-country trial. Details on blinding need to be added (who will be blind to the allocation status of the participants).	Thank you. We have now clarified that the education authorities from the respective cities will provide the sampling frames, and that selection will be done by experts from WHO. Furthermore, we have added a sentence to the section on 'Data management and analysis plan' regarding blinding.
Intervention: The authors have mentioned that the intervention will be co-designed and components of the proposed interventions will be prioritized after the needs assessment phase; the description of the intervention lacks necessary information, e.g., duration of the intervention, who will be the delivery agents, how fidelity of the intervention across sites will be ensured, etc.	Thank you. We have now added more detail to the section 'Interventions' and 'Monitoring and evaluation' to clarify these questions.
Control condition: The description of control arms needs to be added. Considering it is a multi-site trial, treatment as usual for adolescents attending schools across all sites may depend on the context. A brief description of the control condition could facilitate seeing how the intervention will compare across sites.	Thank you. We have added more information on the control arm to the section 'Monitoring and evaluation'.
Primary and secondary outcomes need to be specified as outlined in the trial registry.	Thank you for this comment. To clarify primary outcomes that are in line with what is included in the trial registry, we have now added the word 'primary' to the text and the heading of table 2. What is included in the trial registry as 'Other Outcome Measures' (there was no space for process indicators) is included in the manuscript as 'process indicators', since in our view, this term more accurately reflects what is being measured with the G-SHPPS. We have now also added a paragraph on 'Additional measures' where we describe secondary outcomes.
Participant's two-year follow-up: A brief description of the schools enrolled in the trial needs to be added, e.g., whether these are 10th or 12th-grade schools. Realizing all those participants who will be aged 17 at the time of enrolment will turn 19 years old at the time of primary end-point follow-up and might join the college by the time follow-up assessment will be conducted? A brief description of how all those participants who might have left the schools during the study will be followed up at the time of the primary outcome needs to be added.	Thank you for this comment. As per the section 'Monitoring and evaluation', rather than following up each individual, at the time of follow up, new classes will be selected from the same schools to ensure participation of 13-17 year old students as at baseline. To clarify this point further, we have added additional explanation to the section 'Monitoring and evaluation' and to the section 'Study design and locations'.

Participants' safety: Please address any potential risks or benefits to participants and how they will be mitigated or maximized, respectively, during their participation in the trial across sites. Also, please explain whether participants, especially young adolescents, will be compensated for their time spent co-designing workshops.	Thank you. We have added a paragraph with relevant content to the section on 'Ethics and dissemination'.
Trial governance: Please add a brief description of the trial government or trial management body that will provide oversight to a multi-site trial throughout the duration of the trial.	Thank you. We have added a sentence on the trial management body to the section 'Trial status'.